## Replications

psychology

alerting, endogenous, exogenous, speed-accuracy trade-off

**Author for correspondence:**
C. R. McCormick
e-mail: colin.mccormick@dal.ca

# On the selection of endogenous and exogenous signals

C. R. McCormick[1], R. S. Redden[1], A. J. Hurst[2] and R. M. Klein[1]

[1]Psychology and Neuroscience, Dalhousie University, Halifax, Nova Scotia, Canada
[2]University of Waterloo, Waterloo, Ontario, Canada

CRM, 0000-0001-7326-2560

Alerting is one of the three components of attention which involves the eliciting and maintenance of arousal. A seminal study by Posner *et al.* (Posner MI, Klein R, Summers J, Buggie S. 1973 *Mem. Cognit.* **1**, 2–12 (doi:10.3758/BF03198062)) focused on how changing the interval between an alerting signal and a target would impact the speed and accuracy of responding. Participants indicated whether targets were presented on the left or right side of the fixation point. Auditory warning signals were played at various intervals prior to the target to alert participants and prepare them to make a response. Reaction times revealed a robust, U-shaped, preparation function. Importantly, a clear speed-accuracy trade-off (SAT) was observed. In the current experiment, we replicated the methodological components of this seminal study while implementing a novel auditory warning signal (Lawrence MA, Klein RM. 2013 *J. Exp. Psychol. General* **142**, 560 (doi:10.1037/a0029023)) that was either purely endogenous (change in quality without a change in intensity; analogous to isoluminant colour change in vision) or a combination of endogenous and exogenous (change in both quality and intensity). We expected to replicate the U-shaped preparation function and SAT observed by Posner and colleagues. Based on Lawrence and Klein's findings we also expected the SAT to be more robust with the intense signal in comparison to the isointense signal.

## 1. On the selection of endogenous and exogenous signals

Alerting, which involves the maintenance of arousal and an increase in sensitivity to stimuli, is one of the three networks in Posner's widely used model of attention [1]. In his 1975 *Handbook of Psychobiology*, Posner outlines that tonic and phasic processes differently contribute to alerting. Tonic alerting is more internally guided and reflects continual variations in arousal along with biological processes like our circadian rhythms [2]. Phasic alerting is categorized as rapid physiological

changes in the body in response to external events in order to increase response-readiness [3]. Phasic alerting can be elicited in an individual with relative ease within an experimental context using a sudden visual or auditory event, which we refer to as a signal. The presentation of signals during a trial typically result in improved reaction times (RTs) and decreased accuracy [1]. There are several qualitative factors that contribute to variations in performance, such as the intensity of the signal, probability that the trial will require a response, and the foreperiod length between the signal and response (for a review, see [4]).

In 1971, Posner and Boies attempted to separate the different components of attention in a behavioural task to analyse how each component contributes to human performance [5]. They defined these components as alertness, selectivity, and processing capacity. As mentioned by Posner and Boies, these were preliminary categorizations and years later were refined into the alerting, orienting, and executive functioning networks [1]. Posner and Boies used a letter matching task to dissociate alertness and selectivity. As a way to isolate alertness, they focused on the RT improvement when presenting an auditory signal at different foreperiods before the second letter was presented. To isolate the selectivity stage of processing, they looked at the RT improvement when presenting the first cued letter at different foreperiods before the second letter. In their task, using an alerting signal equally improved RT both when the letters matched and when the letters did not match. When using a selectivity cue or presenting the first letter at varying intervals before the second letter, RT was improved for matched pairs relative to unmatched pairs. When these two qualities were combined, or when a single item was used both to alert and select, the improvements in RT were the sum of the two effects. Posner and Boies concluded by indicating that because these effects were additive, the two processes independently contribute to performance on the task [6].

Following up on Posner and Boies results and conclusion, Posner *et al.* [7] analysed the number of errors generated across the different experiments in their results. This was done to view the relationship between error rate (ER) and RT across the different tasks. They found that although alerting the participants decreased RT, ER remained consistent. This result suggested that the build-up of information obtained through their selectivity cues was rapid, and an individual could make responses faster through alerting signals with pure improvements in performance. While this was 'encouraging for our view of alertness', Posner *et al.* [7, p. 3] qualify that the overall ER was relatively low, making this an unreliable conclusion.

This motivated a follow-up task that was difficult enough to provide a sufficient number of errors for analysis while assessing the relationship between RT and ER. To accomplish this, Posner *et al.* used a localization task with a spatial compatibility manipulation. Participants were required to make spatial responses to left and right targets. Before each block, the participant was informed of the current compatibility condition. A compatible block meant that participants should make a spatially compatible response (ex: left target = left button press). An incompatible block meant that participants should make a spatially incompatible response (ex: left target = right button press). The compatibility condition was maintained in successive blocks of 20 trials. To observe the effect of alerting on performance, they presented a 50 ms tone at five different foreperiod lengths: 50 ms, 100 ms, 200 ms, 400 ms and 800 ms. There was additionally a no-signal condition that was equiprobable, which they, albeit misleadingly, refer to as a 0 ms foreperiod. In doing this, they created a more challenging task that allowed them to look at RT and ER as a function of foreperiod.

They accomplished their methodological goal of increasing the ER (figure 1). What they found, contrary to Posner and Boies, was that the number of errors increased as RT decreased across the foreperiods, generating a relationship known as a speed-accuracy trade-off (SAT) [8,9]. This seminal paper provided the field with a better understanding of how alerting impacted human performance.

An important distinction in the study of alerting that has since generated interest is the distinction of whether it is endogenously or exogenously elicited. Endogenous alerting is a volitionally guided mechanism and can be referred to as a top-down process, as participants rely on some event-related contingency to bring about increased arousal. Exogenous alerting is more reflexive and can be better described as a bottom-up process, as a salient, or 'intense', stimuli in the environment elicit the alerting [10]. While both forms of alerting still provide increases in arousal, they differently impact performance [11] and appear to be independent of each other [12]. Lawrence & Klein [11] developed a methodology to elicit purely endogenous and exogenous forms alerting. This was done using two injunctions.

(i) Intensity: continuous mono (a.k.a diotic) white noise was presented throughout a trial so that a stereo (a.k.a dichotic) signal of uncorrelated white noise could be presented to both ears. This allows for an easy to detect signal regardless of whether the intensity increases (+Δ dB) or remains the same (0Δ dB).

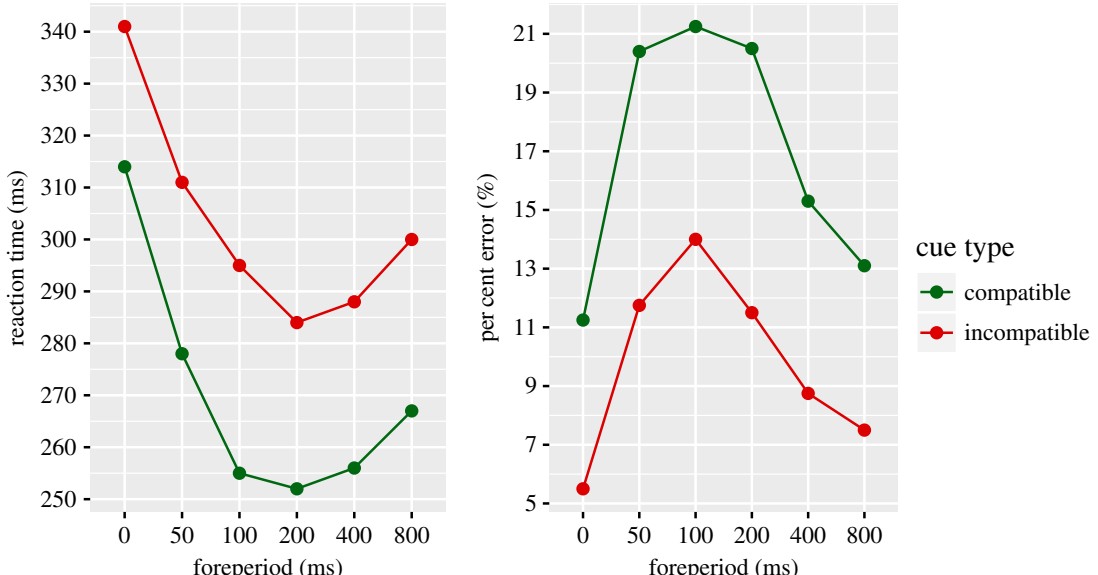

**Figure 1.** Redrawn from Posner *et al*. [7]. To the left is reaction time as a function of stimulus onset asynchrony (SOA). On the right is the error rate as a function of SOA. Compatible and incompatible trials are represented as green or red dots, respectively.

(ii) Contingency: contingency refers to the relationship between the signal and target in an experiment. If the presentation of a signal reliably predicts the presentation of a target, such that the signal and target alternate presentation, it is a contingent (C) design. If the signal and target are randomly presented and have no predictable relationship, it is a non-contingent design (NC). This allows for the control over the learned relationship between signal-target presentation.

Through the combination of these two injunctions, Lawrence & Klein [11] were able to compare performance under these different forms of alerting. In their experiment, participants were required to discriminate whether the presented target was blue or orange by pressing one of two buttons. Participants were put into conditions with different combinations of the two above injunction conditions. If a signal has an intensity increase (+Δ dB) in an NC, it is a *purely exogenous* form of alerting, as it is elicited with a salient presentation and has no influence of a learned contingency. If an there is a no-intensity signal (0Δ dB) and a C design, it is a *purely endogenous* form of alerting, as the alerting is generated solely on the learned contingency. If there is a signal that increases in intensity (+Δ dB) in a C design, this is a *combination* of both endogenous and exogenous alerting, as there is a reflexive response to the intense signal itself, and a learned contingency. Typically, studies manipulating alerting (unknowingly) implement a combined alerting procedure, such as Posner *et al*. [7]. Lawrence and Klein looked at the relationship between RT and ER when using these different combinations of contingency and intensity in their task [11]. In the purely exogenous condition, they saw a brief improvement in RT. In their purely endogenous condition, Lawrence and Klein observed pure improvements in participant performance, with benefits to both speed and accuracy. In their combined alerting condition, they observed a SAT, as would be predicted from Posner *et al*. [7]. These results suggest that these different forms of alerting can be isolated from each other but may also interact.

Our current experiment aims to implement the methodological advances in signalling (as contributed by Lawrence and Klein [11]) within Posner *et al*. [7] seminal alerting experiment design. We presented two-different signal types between-participants in a contingent-only signal-target design: a signal that increases in intensity, which elicits a combination of endogenous and exogenous alerting, and a signal that remains isointense through the use of stereo signalling, which elicits purely endogenous alerting. These two conditions will be referred to as the *combined* condition and the *purely endogenous* condition, respectively. These signals will be used to inform the participant of an upcoming target and will help us measure how these different alerting signals impact performance. This will provide us with two research opportunities.

(i) We will be conducting a long-overdue replication of Posner *et al*. [7] using our combined condition. This will match all the essential methodological components of their original study.

(ii) In addition, we will be comparing performance in the purely endogenous condition to the combined condition to see if there are differences in the relationship between RT and ER, as predicted by Lawrence & Klein [11].

For the replication condition (*combined* condition), we expect to observe an SAT as a consequence of the high-intensity signal relative to the no warning signal stimulus onset asynchrony (SOA; analogous to foreperiod) condition [7,11]. In our purely endogenous condition, we expect to observe genuine improvements to performance as a consequence of the no-intensity warning signal relative to no warning signal SOA condition [11].

# 2. Methods

## 2.1. Registration

This article received results-blind in-principle acceptance (IPA) at Royal Society Open Science. Following IPA, the accepted Stage 1 version of the manuscript, not including results and discussion, was preregistered on the Open Science Framework (OSF). This preregistration was performed after data analysis. All further project materials, including experiment code and participant data, are also located on this project's OSF page (https://osf.io/ry9vm/).

## 2.2. Participants

Participants were run until we had 24 useable participants in both alerting conditions, resulting in 48 participants total.[1] Useable is defined as having completed the task as directed, specifically having responded to 75% of trials and an overall accuracy rate above chance.

## 2.3. Materials

All stimuli for this experiment were presented on 21.5″ Apple iMac computers running OS X 10.9.5 in a group testing room at Dalhousie University. Visual stimuli were displayed at a resolution of $1920 \times 1080$ pixels with a refresh rate of 60 Hz. Audio stimuli were presented at a sample rate of 44 100 Hz using Sony MDR- 101LP headphones attached to the headphone ports of the computers. Responses were collected using Apple USB keyboards (model A1243).

The experiment program for this study was written in Python using the KLibs framework for cognitive psychology experiments. The experiment code, along with an animation illustrating several trials of the task and instructions on how to download and replicate the study, can be found on our laboratories GitHub page (https://github.com/TheKleinLab/TaskSwitching) [13].

The stimuli presented at each stage of the trial are illustrated in figure 2. All stimulus sizes are defined in terms of their perceptual size to participants in degrees of visual angle (°). All text in the experiment was displayed in 28pt (0.5° tall) Frutiger typeface. Throughout the experiment, unless otherwise stated, a single channel of randomly generated uniform white noise was presented to both ears via the headphones at a set volume.

At the onset of each trial (figure 2a), a white fixation ring (1.0° diameter, 0.4° inner stroke) was displayed in the middle of a black background, flanked on the left and right by white target placeholder boxes (1.0° size, 0.1° inner stroke, 11.1° horizontal offset from fixation). Surrounding these stimuli was a border cue (5.0° from all edges of the screen, 0.1° inner stroke) that was either green or red, indicating whether participants would have to make a compatible response (green) or an incompatible response (red) to the target on that trial (explained in greater detail below).

## 2.4. Procedure

Participants completed four blocks of 64 trials each. Participants were provided with the opportunity for a break halfway through the task. Each block consisted of four sets of 16 trials, one for each of the four possible signal delay conditions[2] (no signal, 100, 250 and 850 ms), with the order of the sets being

---

[1]This satisfies Simonsohn's [14] criterion of 2.5 times the original sample size (Posner *et al.* n = 9) to detect an effect the same size or larger.

[2]Note that these SOAs were meant to be a subset of the SOAs that Posner *et al.* [7] used (0, 50, 200, and 800). This was to ensure we could see the 'U-shaped' function of the original study. When piloting our experiment, we realized that the no intensity signal was

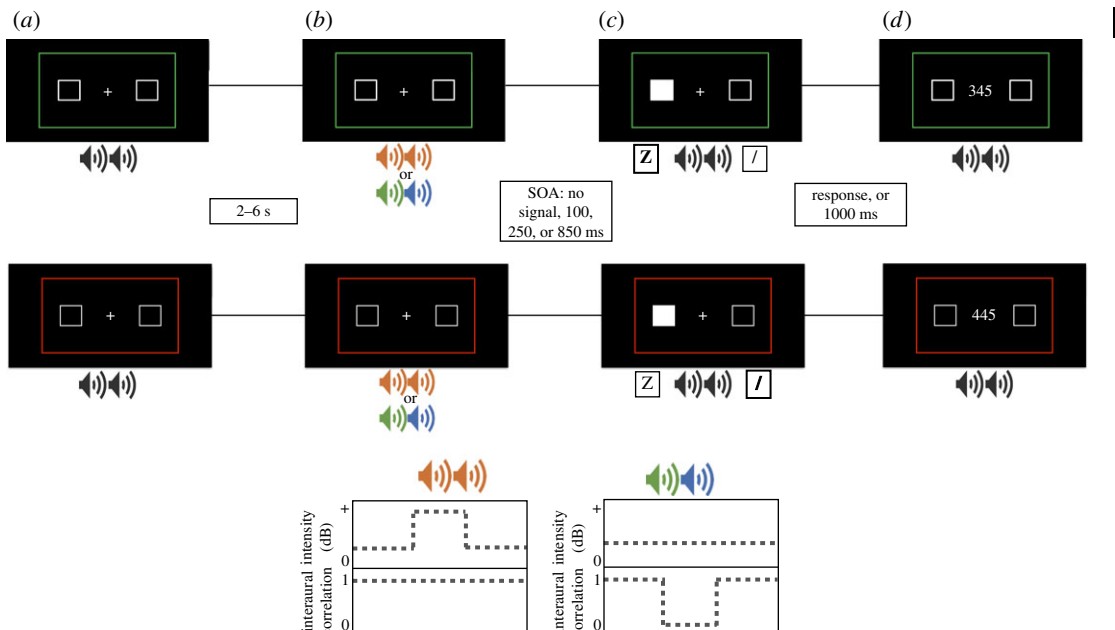

**Figure 2.** A breakdown of a trial in the current experiment. Mono white noise is played through the participant's headphones throughout the task. A compatible trial, indicated by a green border, is displayed on the top sequence, while an incompatible trial, displayed by a red border, is displayed on the bottom sequence. (*a*) Participants are instructed to fixate on the centre cross. A random interval between 2 and 6 s occurs to minimize a participant's ability to anticipate when a signal would be presented. (*b*) A signalling tone is played to alert the participant that there is an upcoming target stimulus. This signal either increases in intensity (red sound image; left) or remains the same intensity and uses uncorrelated stereo-sound (blue-green sound image; right), depending on the signal condition. (*c*) After one of the possible SOA 'conditions' (no signal, 100, 250 or 850 ms), the target is presented by filling in one of the possible placeholder boxes. On a compatible trial, this requires a button press on the same side as the target, while on an incompatible trial this requires a button press on the opposite side of where the target is. (*d*) If a correct response is made, feedback is presented as three numbers which represent the participants RT in milliseconds. If they are incorrect, or take longer than 1000 ms, an error message is displayed.

random within each block. The colour of the border cue was chosen randomly at the start of each session and alternated every eight trials throughout the experiment, resulting in eight 'compatible' and eight 'incompatible' trials for each signal delay condition in each block (see figure 3 for an illustration).

Following trial onset by a random interval between 2000 and 6000 ms (figure 2*a*), a brief (100 ms) auditory alerting signal was presented on 75% of all trials (figure 2*b*). For participants in the exogenous alerting condition, this was a temporary increase in noise volume to 100%. For participants in the endogenous alerting condition, this was the temporary presentation of different streams of randomly generated noise to the left and right ears with no change in intensity. On the 25% of trials without an alerting signal, the target for the trial was presented immediately after the end of the onset interval. On the 75% of trials with an alerting signal, the presentation of the target was either 100, 250 or 850 ms after the onset of the signal (figure 2*c*).

Once the target was presented, participants were given 1000 ms to make a response by pressing either the 'z' or '/' keys on the keyboard to indicate 'left' and 'right', respectively. Participants were told to respond as quickly as possible, but that accuracy was still important. On trials with a green border cue (i.e. compatible trials), participants were required to press the key on the same side as the target. On trials with a red border cue (i.e. incompatible trials), participants were required to press the key on the side opposite to the target. As soon as a response was made, the fixation ring was replaced with feedback on trial performance for a period 1000 ms. If a correct response was made, the speed of the response (in milliseconds) was displayed in place of the ring (figure 2*d*). If an incorrect response

challenging to detect, so we doubled the signal length from 50 ms to 100 ms. At the new signal length, our pilot participants had no difficulty detecting its presence, and so we added 50 ms to each SOA. This ensured that the length of time after the signal offset-target onset matched that of the original study, and so that in the 100 SOA condition, the signal had completed playing before the target was presented.

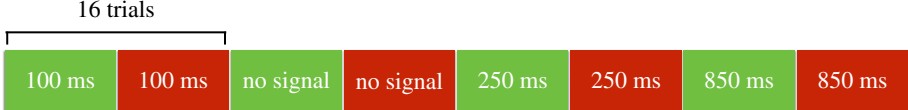

**Figure 3.** An illustration of the structure of a possible block from the experiment. Green blocks represent sets of eight 'compatible' trials, red blocks represent sets of eight 'incompatible' trials. The text indicates the SOA condition for each set of trials.

was made, a white 'X' (a 1.0° cross rotated 45 degrees, 0.1° thick) was displayed instead. If no response was detected, the text 'Too Slow!' was displayed at fixation.

# 3. Results

## 3.1. Data processing

Before analysing the data, we trimmed both ends of the RT distribution. This was done by binning responses by RT ranges (example: 0–50 ms, 50.1–100 ms, 100.1–150 ms, etc.) and observing the frequency of responses, as well as the average ER. The cut-off is determined when the ER for the bin sufficiently changes above chance, in addition to considering the frequency of responses that fall within that bin. At the faster end of the distribution, this procedure affords an objective measure for removing anticipatory responses, as there is a clear improvement in accuracy. At the slower end of the response distribution, we can determine when there is an increase in task unrelated responses based on a dip in accuracy. Once there were bins that were identified as significantly deviating from the bin preceding, considering the frequency of responses and per cent correct, this bin of 50 ms was split into five bins (width = 10 ms) to try to be as precise in excluding data as possible. Following this for the current dataset, responses that were faster (1.2% removed) than 200 ms (84% accuracy and 343 trials in 200–250 ms bin as compared to 150–200 ms bin, with 62% accuracy and 68 trials) and slower (2.1% removed) than 710 ms (90% accuracy and 185 trials in 650–700 ms as compared to 700–750 ms, with 84% accuracy and 94 trials) were excluded for these reasons.[3] Additionally, any misses, or no responses, were removed (1.2% of trials). After making these exclusions, the number of trials contributed by each participant was checked to ensure that there were no participants who lost a substantial amount of their data and needed to be excluded from analyses. The participant with the fewest trials was 221 out of 256 (86.3% remaining, $M = 95.5\%$), so none of the 48 participants were excluded.

## 3.2. Statistical tests

Results from 48 participants (eight male, five left-handed, $M$age = 20.2, age range: 17–30 years) were analysed in v.3.6.0 of the R statistical software [15] using the afex package [16]. To guard against violations of the ANOVA's assumption of sphericity, Greenhouse-Geisser correction was applied to all reported values involving within-subjects factors. An *a priori* significance threshold of $\alpha = 0.05$ was used for all statistical tests. The mean RTs and ERs for all cells of the design are reported in table 1, and are visualized in figure 4.

A 4 (SOA) × 2 (compatibility) × 2 (alerting type) type III mixed-factorial ANOVA was conducted on RTs, revealing a significant main effect of SOA, $F_{2.5,115.12} = 140.09$, mean squared error (MSE) = 725.28, $p < 0.001$, $\eta_G^2 = 0.260$, with a typical U-shaped function showing a minimum at our 250 ms SOA (figure 4). There was also a significant main effect of compatibility, $F_{1,46} = 204.79$, MSE = 635.08, $p < 0.001$, $\eta_G^2 = 0.152$, with compatible responses (369.92 ms) being faster than incompatible responses (406.73 ms). There was no significant effect for alerting type, $F_{1,46} = 0.68$, MSE = 12, 431.69, $p = 0.414$, $\eta_G^2 = 0.012$. There was a significant interaction between alerting type and SOA, $F_{2.5, 115.12} = 3.45$, MSE = 725.28, $p = 0.026$, $\eta_G^2 = 0.009$.

A 4 (SOA) × 2 (compatibility) × 2 (alerting type) type III mixed-factorial ANOVA was conducted on ERs. There was a significant effect of SOA, $F_{2.77, 127.48} = 20.92$, MSE = 0.00, $p < 0.001$, $\eta_G^2 = 0.100$, with an inverted U-shaped function showing a maximum ER at our 100 and 250 ms SOA (figure 4). There was also a significant effect of compatibility, $F_{1,46} = 16.24$, MSE = 0.01, $p < 0.001$, $\eta_G^2 = 0.074$, with

[3]For the more specific 10 ms bin breakdown that led to the specific cut-offs: the 190–200 ms bin had 50% accuracy and 20 trials compared to the 200–210 ms bin with 71% accuracy and 28 trials; 700–710 ms had 91% accuracy and 23 trials as compared to 710–720 ms, with 80% accuracy and 25 trials.

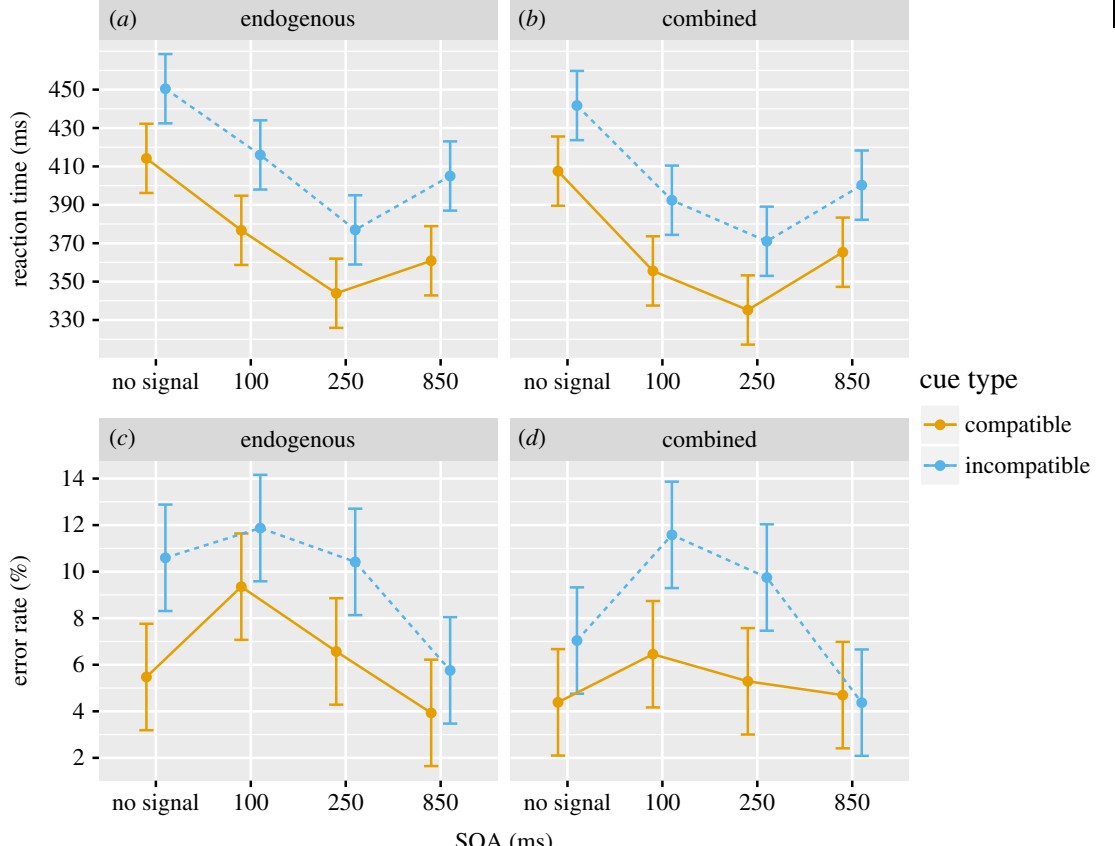

**Figure 4.** Reaction time (*a,b*) and error rate (*c,d*) across the different SOAs for the current experiment. The solid line represents compatible trials, while the dotted line represents incompatible trials. Figures are split by signalling types. Error bars are 95% confidence intervals.

**Table 1.** Mean reaction time and error rate for the various conditions.

|  | reaction time (ms) | | | | error rate (%) | | | |
|---|---|---|---|---|---|---|---|---|
|  | none | 100 ms | 250 ms | 850 ms | none | 100 ms | 250 ms | 850 ms |
| **endogenous** | | | | | | | | |
| compatible | 414.20 | 376.71 | 343.93 | 360.85 | 5.47 | 9.36 | 6.57 | 3.93 |
| incompatible | 450.49 | 415.99 | 376.95 | 405.01 | 10.60 | 11.87 | 10.42 | 5.76 |
| **combined** | | | | | | | | |
| compatible | 407.57 | 355.57 | 335.21 | 365.31 | 4.39 | 6.45 | 5.29 | 4.70 |
| incompatible | 441.72 | 392.39 | 371.02 | 400.25 | 7.04 | 11.58 | 9.75 | 4.37 |

compatible responses (5.77%) being more accurate than incompatible responses (8.92%). There was no significant effect for alerting type, $F_{1,46} = 2.19$, MSE = 0.01, $p = 0.145$, $\eta_G^2 = 0.013$. There was a significant interaction between compatibility and SOA, $F_{2.93,134.92} = 2.98$, MSE = 0.00, $p = 0.035$, $\eta_G^2 = 0.015$.

## 4. Discussion

This experiment had two main research objectives. First, the paradigm was designed to replicate the first experiment from Posner *et al*. [7], in which they reported a SAT in an alerting task across various SOA conditions. This replication effort is implemented in our combined alerting condition. Second, a novel

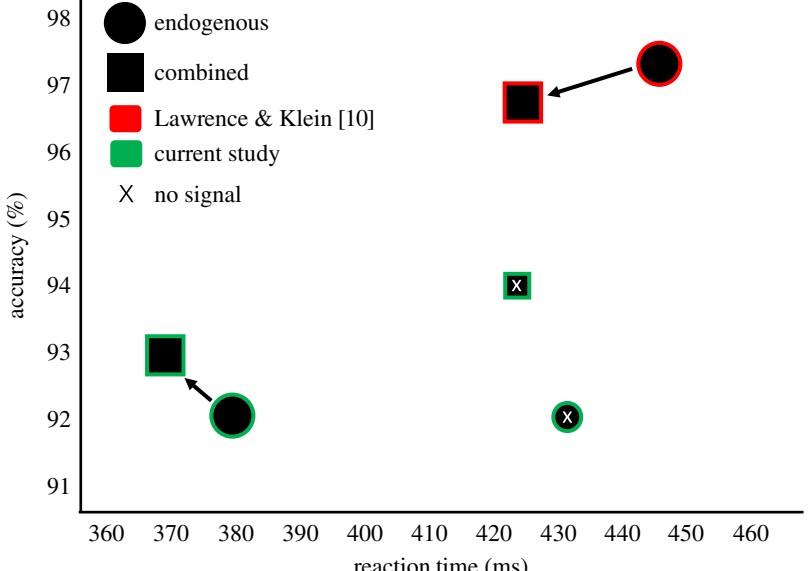

**Figure 5.** Mapping the endogenous (circle) and combined (square) alerting conditions in speed-accuracy space. Both the current experiment (green) and Lawrence and Klein (red) are represented to compare the current results to the results of their study, which the current experiments hypotheses were based on. The smaller shapes with the x's are no-signal means, while the larger shapes are the three other SOA condition means collapsed together.

manipulation was added to the task that allowed for the elicitation of a purely endogenous form of alerting. Based on Lawrence & Klein [11], genuine improvements to performance were predicted as a consequence of the purely endogenous signal type relative to the no signal SOA condition, in which improvements in both RT and ER would be observed.

The results from the combined signal condition indicate an overall successful replication of Posner *et al*. We observe a U-shaped function for RT that is similar to what was reported in the original paper: the fastest mean response occurred when the signal was presented 250 ms before the target. We additionally observe a similar inverted U-shaped function for ER, in which the greatest number of errors were generated in the 100 ms and 250 ms SOA conditions, corresponding to the instances when responses are fastest. When comparing the current results (figure 4) to Posner *et al*.'s original findings (figure 1), it is evident that the important features of performance have been replicated. We found a significant interaction between compatibility condition and SOA for ER, which appears to be driven by lack of differences between SOAs in the compatible condition. This is worth mentioning as the compatible condition does not precisely follow the expected pattern observed in the original study. This could be attributed to the speed at which participants responded, as our mean response time is significantly slower than Poser *et al*.'s. This speed difference could be owing to experimental factors related to the 45-year gap in when these studies were conducted, like changes in the equipment used or cohort effects. A reduction in speed could result in decreased magnitude of accuracy differences across the SOA conditions in our experiment.

The purely endogenous condition was compared to the combined condition to examine whether there would be a significant difference in RT and ER performance, as predicted by Lawrence & Klein [11]. There were no significant differences for RT or ER. This can be seen when visually inspecting the data (figure 4), as both of these figures produce near-identical SAT functions. Additionally, when looking at the mean 'no signal' performance relative to the mean 'signalled' performance (all other SOAs) in SAT space (figure 5), the genuine improvement in performance (lower RT and decreased ER) as predicted by Lawrence & Klein [11] was not observed. Additionally, the positioning of these two signalling types relative to one another does not match Lawrence and Klein's past finding (figure 5). When considering the differences between these two signalling types, in the face of failure to produce the expected contrasts, a particular pattern worth mentioning is the interaction between alerting type and SOA. It appears that these two alerting conditions differed in the time-course of RT performance, specifically at 100 ms, with the combined alerting condition being significantly faster than the endogenous alerting condition. This difference dissipates at the 250 ms SOA, in which both conditions

reach similar peak RT performance. This early-advantage for RT in the combined alerting condition in comparison to the purely endogenous condition is also found in Lawrence and Klein's results [11]. So, although this time-course result does represent a previously-observed difference between these two signalling types, the expected RT and ER relationship that would be predicted for the purely endogenous condition does not represent the previous literature. Instead, it matches what would be predicted, and what was found, for the combined alerting condition [11].

There is some evidence that these different modes of alerting are impacting the tonic response state of participants, aside from the phasic effects that were of central interest. This is suggested by examination of performance in the no-signal conditions (figure 5). The no-signal trials are particularly analytic, as on these trials both signalling groups experience identical perceptual conditions. Upon comparison, participants experiencing the purely endogenous 'isointense' signal during the block were less accurate than those experiencing the combined 'intense' signal (while there was no difference in RT) when there was no signal presented. This suggests that exposure to these two different signal types over a block of trials affects the participant's overall mental state of responding. This tonic effect on performance could be related to the increased cognitive demand that the purely endogenous signal has been theorized to require over the intense combined signal, so blocks with these isointense signals would require additional perceptual resources for detection [12]. Future research should investigate how these different alerting types impact tonic levels of arousal, and how the isointense signal used in the endogenous condition impacts cognitive load.

In summary, this experiment successfully replicated the seminal results of Posner *et al.* [7]. This reinforces the theory that alerting increases response speed without improving the quality of information processed about the stimulus of interest, changing the point at which information is consulted to generate a decision. It should be noted that since the publication of Posner *et al.*'s paper [7], it has been argued that the type of alerting elicited by their signal is more specifically defined as a combined type, involving both endogenous and exogenous modes of alerting. Additionally, there was a failure to reproduce the effects on performance that would be expected by Lawrence & Klein [11]. The different alerting conditions defined in Lawrence and Klein should be further studied under various methodological conditions and domains to examine the generalizability of their theory, as is outlined in their proposed taxonomy for studying attention [17].

Ethics. Ethics for this experiment were obtained through Dalhousie's Research Ethics Board (REB no. 2018-4505).

Data accessibility. All experimental materials and data can be accessed through going to: https://github.com/TheKleinLab/TaskSwitching.

Authors' contributions. C.R.M. wrote the manuscript, ran the participants through the task, and contributed to data analysis. R.S.R. contributed intellectually to the revisions of the manuscript and assisted with data analysis. A.J.H. wrote the experimental code, helped with ensuring all the experimental materials and data were open access, and assisted with data analysis. R.M.K. contributed intellectually to all of the steps of this research project and conceived of the study. All authors approve of the current version of the project and welcome questions related to the project.

Competing interests. We declare that we have no competing interests.

Funding. C.R.M. Dalhousie Graduate Fellowship. R.S.R. Killam Trusts Doctoral Award & NSERC Canadian Graduate Scholarship. A.J.H. and R.M.K. NSERC Discovery Grant.

Acknowledgements. We thank Bronwyn O'Connor, Lindsey Puddicombe, Samantha Howard and Piper Sawchyn for their assistance with running participants for this experiment. We also thank our laboratory manager Swasti Arora for her operational support.

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
