## [Reviewer comments · Royal Society Open Science]

Review History

RSOS-190134.R0 (Original submission)

Review form: Reviewer 1

Do you have any ethical concerns with this paper?

No

Have you any concerns about statistical analyses in this paper?

Yes

Recommendation?

Accept with minor revision

Comments to the Author(s)

My main concerns relate to the power and analysis pipeline

1. There is no discussion of how or why a sample of 24 participants was selected. I has expected to see an apriori power calculation based on effects previously observed in the literature used to justify the sample size

2. The analysis doesn't seem to address the hypothesis 2, which is "comparing performance in the high-intensity signal condition to the no-intensity signal condition to see if there are differences in the relationship between RT and ER, as predicted by Lawrence and Klein (2013)".

The 2x2 anova described in the analysis section does not include a high v low intensity factor, so these conditions are not statistically compared. It seems to me that a 2x2x2 omnibus anova would be more appropriate to address hypothesis 2

Review form: Reviewer 2

Do you have any ethical concerns with this paper?

No

Have you any concerns about statistical analyses in this paper?

No

Recommendation?

Accept with minor revision

Comments to the Author(s)

This study implements a relatively recent auditory stimulus technique, which accounts for differences between exogenous and endogenous alerting, and applies it to a well-established paradigm with the intention of replicating the results originally obtained from this paradigm, as well as determining the independent influence of endogenous and exogenous alerting on this established pattern of results.

Generally, the motivations of this study are well structured and clearly laid out, and without a doubt demonstrate the importance of carrying out this research. Most importantly, the description of the upcoming methods and analyses are put forward in a transparent fashion, though there are some discrepancies in the description of stimuli and procedures; hopefully these are due simply to errors in writing. Overall, several issues will need to be addressed before I can recommend this manuscript for publication in principle. These issues are laid out below:

1) When describing Lawrence & Klein's (2013) methodology, the description of the concept of "contingency" is lacking in clarification. Since I have read this paper myself, I understand that contingency in this case means that the order of presentation of signal and target was predictable and alternating. Nowhere in your manuscript is this clarified, and readers may be confused and potentially assume that you are referring to the identity or the location of the signal with respect to the target. Also, clarifying the actual procedure of Lawrence & Klein's experiment will help readers to understand that their task was qualitatively different, and to appreciate the motivation for the current study.

2) The figures you have uploaded are extremely low in DPI quality, to the extent that they are difficult to read. Please increase the DPI substantially before re-uploading.

3) On the 7th page (counting from the first page of the introduction), on line 17, the authors refer to "Figure 1", but are in fact referring to Figure 3.

4) On the 8th page, line 31/32, the authors state that the auditory signal was 50ms. However, in the footnote on the same page, they state that this signal was increased to 100ms. Please ensure this is consistent and correct.

5) On the 9th page, line 3, the authors state that the target presentation followed the *end of the auditory signal by 100/250/850ms*. However, these are SOA times (stimulus *onset* asynchrony), and the reason for the change in SOA compared to the original study was to ensure that the time between the *end* of the auditory signal and the onset of the target was in fact identical to the previous study. Therefore, I strongly suspect that the authors actually mean to say that the time between the *start* of the auditory signal and the onset of the target was 100/250/850ms. Please revise this section and ensure all descriptions of stimulus timing are correct.

6) It is not clear why the SOA time needed to be increased simply because the auditory signal length increased; there is nothing special about the time between the end of the auditory signal and the start of the target onset; participants can begin to respond to the auditory stimulus as soon as it is perceived, which is why stimulus onset asynchrony is considered a more important metric. In my opinion, it would be better to keep the original SOA times unchanged.

7) The use of past and future tense throughout the methods section is inconsistent, e.g., "participants will be run..." vs. "participants completed four blocks...". Please ensure you comply with the publication guidelines and keep the writing to past tense at this stage.

8) Finally, there are a very large number of writing errors and typos (e.g., the word "stimuli" is used for singular, when it should only be used for plural; "stimulus" is the singular term), as well as formatting errors (e.g., 5th page contains a sentence in a different typeface). These errors appear throughout, but are especially severe in the Figure legend of Figure 3. The authors should carefully review all sections of the manuscript to ensure no spelling or grammatical errors.

Provided the authors substantially reduce their grammatical/spelling errors, and add some additional clarifications in their introduction (particularly regarding the SOA variations relative to the original Posner study), I will be happy to accept this manuscript for publication in principle.

Review form: Reviewer 3

Do you have any ethical concerns with this paper?

No

Have you any concerns about statistical analyses in this paper?

No

Recommendation?

Accept with minor revision

Comments to the Author(s)

This study was designed to replicate the seminal study by Posner et al, 1973, with both endogenous and exogenous signals adopted in Lawrence and Klein, 2013. The authors, in doing this way, tried to explore the difference on speed-accuracy tradeoff between endogenous and exogenous signals.

The logic behind this study is very straightforward. This kind of comparison between endogenous and exogenous signals would have broad audiences in the field of attention. It also has strong general interesting. I only have some suggestions on the experimental design that might help improving the paper and make it more accessible to a wider audience.

1, Based on the definition of different alerting signals in Lawrence and Klein (2013), and the logic behind the current study, I would like to see a fully design, i.e., 2 (Contingency: yes vs. no) * 2 (Intensity: change vs. no-change) * 4 (SOAs: 0, 50, 200, and 800 ms) * 2 (Compatibility: yes vs. no). I believed that, only in this way, the authors could compare endogenous (0 Δ dB and C) and exogenous signals (+ Δ dB and NC). Otherwise, as described in the paper that there was no NC condition, we could only examine the difference between endogenous signal (0 Δ dB and C) and both endogenous and exogenous signal (+ Δ dB and C).

2, As illustrated in Figure 1, 200 ms is the peak for RT, and 100 ms is the peak for accuracy. I think it is valuable to involve both in.

Minor points

- 1, See p. 9. In the main text, the authors mentioned Figure 1 and 1a, but I cannot find them in Figure 1. Please check it.
- 2, Figure 2 is hard to be understood. Please give more explanation, or change it.
- 3, In Figure 3, I think it would be helpful for reader to understand, if you also put SOA 0 ms on it.
- 4, Figure 3 caption, line 39. 'A interval' - 'An interval'.

Decision letter (RSOS-190134.R0)

25-Feb-2019

Dear Mr McCormick

On behalf of the Editors, I am pleased to inform you that your Stage 1 Replication RSOS-190134 entitled "On the Endogenous and Exogenous Selection of Signals" deemed suitable for in-principle acceptance in Royal Society Open Science subject to minor revision in accordance with the referee and editor suggestions. Please find their comments at the end of this email.

The reviewers and handling editors have recommended publication, but also suggest some minor revisions to your manuscript. Therefore, I invite you to respond to the comments and revise your manuscript.

When submitting your revised manuscript, you will be able to respond to the comments made by the referees and upload a file "Response to Referees" in the "File Upload" step. You can use this to document any changes you make to the original manuscript. In order to expedite the processing of the revised manuscript, please be as specific as possible in your response to the referees.

Full author guidelines can be found here

<http://rsos.royalsocietypublishing.org/page/replication-studies#AuthorsGuidance>.

Kind regards,

Professor Chris Chambers

on behalf of Professor Chris Chambers (Registered Reports Editor, Royal Society Open Science)

Associate Editor Comments to Author (Professor Chris Chambers):

Three expert reviewers have appraised the manuscript. As you will see, all find merit in this replication study but also note a number of areas in the rationale and methods requiring greater clarity and justification, including most notably the sample size and mapping between the hypotheses and analyses. In revising the manuscript, please attend carefully to every point raised in the reviews.

Reviewers' comments to Author:

Reviewer: 1

Comments to the Author(s)

My main concerns relate to the power and analysis pipeline

1. There is no discussion of how or why a sample of 24 participants was selected. I have expected to see an a priori power calculation based on effects previously observed in the literature used to justify the sample size

2. The analysis doesn't seem to address the hypothesis 2, which is "comparing performance in the high-intensity signal condition to the no-intensity signal condition to see if there are differences in the relationship between RT and ER, as predicted by Lawrence and Klein (2013)".

The 2x2 ANOVA described in the analysis section does not include a high v low intensity factor, so these conditions are not statistically compared. It seems to me that a 2x2x2 omnibus ANOVA would be more appropriate to address hypothesis 2

Reviewer: 2

Comments to the Author(s)

This study implements a relatively recent auditory stimulus technique, which accounts for differences between exogenous and endogenous alerting, and applies it to a well-established paradigm with the intention of replicating the results originally obtained from this paradigm, as well as determining the independent influence of endogenous and exogenous alerting on this established pattern of results.

Generally, the motivations of this study are well structured and clearly laid out, and without a doubt demonstrate the importance of carrying out this research. Most importantly, the description of the upcoming methods and analyses are put forward in a transparent fashion,

though there are some discrepancies in the description of stimuli and procedures; hopefully these are due simply to errors in writing. Overall, several issues will need to be addressed before I can recommend this manuscript for publication in principle. These issues are laid out below:

1) When describing Lawrence & Klein's (2013) methodology, the description of the concept of "contingency" is lacking in clarification. Since I have read this paper myself, I understand that contingency in this case means that the order of presentation of signal and target was predictable and alternating. Nowhere in your manuscript is this clarified, and readers may be confused and potentially assume that you are referring to the identity or the location of the signal with respect to the target. Also, clarifying the actual procedure of Lawrence & Klein's experiment will help readers to understand that their task was qualitatively different, and to appreciate the motivation for the current study.

2) The figures you have uploaded are extremely low in DPI quality, to the extent that they are difficult to read. Please increase the DPI substantially before re-uploading.

3) On the 7th page (counting from the first page of the introduction), on line 17, the authors refer to "Figure 1", but are in fact referring to Figure 3.

4) On the 8th page, line 31/32, the authors state that the auditory signal was 50ms. However, in the footnote on the same page, they state that this signal was increased to 100ms. Please ensure this is consistent and correct.

5) On the 9th page, line 3, the authors state that the target presentation followed the *end of the auditory signal by 100/250/850ms. However, these are SOA times (stimulus onset asynchrony)*, and the reason for the change in SOA compared to the original study was to ensure that the time between the *end* of the auditory signal and the onset of the target was in fact identical to the previous study. Therefore, I strongly suspect that the authors actually mean to say that the time between the *start* of the auditory signal and the onset of the target was 100/250/850ms. Please revise this section and ensure all descriptions of stimulus timing are correct.

6) It is not clear why the SOA time needed to be increased simply because the auditory signal length increased; there is nothing special about the time between the end of the auditory signal and the start of the target onset; participants can begin to respond to the auditory stimulus as soon as it is perceived, which is why stimulus onset asynchrony is considered a more important metric. In my opinion, it would be better to keep the original SOA times unchanged.

7) The use of past and future tense throughout the methods section is inconsistent, e.g., "participants will be run..." vs. "participants completed four blocks...". Please ensure you comply with the publication guidelines and keep the writing to past tense at this stage.

8) Finally, there are a very large number of writing errors and typos (e.g., the word "stimuli" is used for singular, when it should only be used for plural; "stimulus" is the singular term), as well as formatting errors (e.g., 5th page contains a sentence in a different typeface). These errors appear throughout, but are especially severe in the Figure legend of Figure 3. The authors should carefully review all sections of the manuscript to ensure no spelling or grammatical errors.

Provided the authors substantially reduce their grammatical/spelling errors, and add some additional clarifications in their introduction (particularly regarding the SOA variations relative to the original Posner study), I will be happy to accept this manuscript for publication in principle.

Reviewer: 3

Comments to the Author(s)

This study was designed to replicate the seminal study by Posner et al, 1973, with both endogenous and exogenous signals adopted in Lawrence and Klein, 2013. The authors, in doing this way, tried to explore the difference on speed-accuracy tradeoff between endogenous and exogenous signals.

The logic behind this study is very straightforward. This kind of comparison between endogenous and exogenous signals would have broad audiences in the field of attention. It also has strong general interesting. I only have some suggestions on the experimental design that might help improving the paper and make it more accessible to a wider audience.

1, Based on the definition of different alerting signals in Lawrence and Klein (2013), and the logic behind the current study, I would like to see a fully design, i.e., 2 (Contingency: yes vs. no) * 2 (Intensity: change vs. no-change) * 4 (SOAs: 0, 50, 200, and 800 ms) * 2 (Compatibility: yes vs. no). I believed that, only in this way, the authors could compare endogenous (0 Δ dB and C) and exogenous signals (+ Δ dB and NC). Otherwise, as described in the paper that there was no NC condition, we could only examine the difference between endogenous signal (0 Δ dB and C) and both endogenous and exogenous signal (+ Δ dB and C).

2, As illustrated in Figure 1, 200 ms is the peak for RT, and 100 ms is the peak for accuracy. I think it is valuable to involve both in.

Minor points

1, See p. 9. In the main text, the authors mentioned Figure 1 and 1a, but I cannot find them in Figure 1. Please check it.

2, Figure 2 is hard to be understood. Please give more explanation, or change it.

3, In Figure 3, I think it would be helpful for reader to understand, if you also put SOA 0 ms on it.

4, Figure 3 caption, line 39. 'A interval' - 'An interval'.

Author's Response to Decision Letter for (RSOS-190134.R0)

See Appendix A.

Decision letter (RSOS-190134.R1)

08-May-2019

Dear Mr McCormick

On behalf of the Editor, I am pleased to inform you that your Manuscript RSOS-190134.R1 entitled "On the Endogenous and Exogenous Selection of Signals" has been accepted in principle for publication in Royal Society Open Science.

You may now progress to Stage 2 and complete the study as approved.

Please note that you must now register your approved protocol on the Open Science Framework (<https://osf.io/rr>), using the 'Submit your approved Registered Report' option and then the 'Registered Report Protocol Preregistration' option. Please use the Registered Report option even though your article is being accepted as a Stage 1 Replication. Further into the registration process, in the Journal Title field enter 'Royal Society Open Science (Replication article type, Results-Blind track)'. Please note that a time-stamped, independent registration of the protocol is mandatory under journal policy, and manuscripts that do not conform to this requirement cannot be considered at Stage 2. The protocol should be registered unchanged from its current approved state. Please include a URL to the protocol in your Stage 2 manuscript, and because you submitted via the Results-Blind track please note in the manuscript that the pre-registration was performed after data analysis (e.g. 'This article received results-blind in-principle acceptance (IPA) at Royal Society Open Science. Following IPA, the accepted Stage 1 version of the manuscript, not including results and discussion, was preregistered on the OSF (URL). This preregistration was performed after data analysis.')

We would be grateful if you could now update the journal office as to the anticipated completion date of your study.

Following completion of your study, we invite you to resubmit your paper for peer review as a Stage 2 Replication. Please note that your manuscript can still be rejected for publication at Stage 2 if the Editors consider any of the following conditions to be met:

- The Introduction and methods deviated from the approved Stage 1 submission (required).
- The authors' conclusions were not considered justified given the data.

We encourage you to read the complete guidelines for authors concerning Stage 2 submissions at: <http://rsos.royalsocietypublishing.org/page/replication-studies#AuthorsGuidance>. Please especially note the requirements for data sharing and that withdrawing your manuscript will result in publication of a Withdrawn Registration.

Once again, thank you for submitting your manuscript to Royal Society Open Science and I look forward to receiving your Stage 2 submission. If you have any questions at all, please do not hesitate to get in touch. We look forward to hearing from you shortly with the anticipated submission date for your stage two manuscript.

Kind regards,
Professor Chris Chambers
Royal Society Open Science
openscience@royalsociety.org

on behalf of Professor Chris Chambers (Registered Reports Editor, Royal Society Open Science)
openscience@royalsociety.org

Author's Response to Decision Letter for (RSOS-190134.R1)

See Appendix B.

RSOS-190134.R2 (Revision)

Review form: Reviewer 1

Do you have any ethical concerns with this paper?

No

Have you any concerns about statistical analyses in this paper?

No

Recommendation?

Accept with minor revision

Comments to the Author(s)

Minor:

I found the explanation for the discrepancy between the current results and those of Lawrence & Klein (2013) shown in fig 5 a bit hard to follow. As I understand it, L&K show faster RT but (slightly) reduced accuracy for a combined cue, whereas as the current study shows faster RT and improved accuracy. However, this pattern is also observed in the 'no signal' trials. There is some reference to participants having different mental states in the two studies. Is the idea that something about the design of the current study means that participants are better in the combined condition, irrespective of the presence of an alerting signal, and this is the cause of the inconsistency with L&K?

Final paragraph, 2nd sentence reads "This reinforces the theory that alerting increases response speed without improving the quality of information processed about the stimulus of interest, and instead is consistent with the suggestion that alerting changes the point at which information is consulted to generate a decision."

This is a bit confusing and the 1st & 2nd part of the sentence seem to be offering similar interpretations of the data, but the 'and instead' connector suggests the data are consistent with one interpretation but not the other

Review form: Reviewer 2

Do you have any ethical concerns with this paper?

No

Have you any concerns about statistical analyses in this paper?

No

Recommendation?

Accept with minor revision

Comments to the Author(s)

Thank you for your Stage 2 Submission; it was a pleasure to go through your results. Overall I am satisfied that this submission fulfills all the relevant criteria. Nonetheless, I do have some small points that would need to be addressed, or at the very least considered. Assuming that point 4 is

addressed, I would be happy to accept this without further review (although I would prefer to see points 1 and 3 addressed as well).

1) Page 11, lines 42/43: The authors report a cut-off for trial exclusion as including “a respectable number of trials that fall within that bin”. What is meant by respectable? Was there an objective threshold for this?

2) Page 11, line 36 to page 12, line 27 (“Data Processing” section): This binning method is in a way refreshing, as most studies just use ‘default’ threshold of around 200 and 1800. However, I am not convinced that using changes in accuracy from bin to bin is a logically sound method for trial exclusion. For slow responses, the accuracy (84%) is still very high, suggesting a large proportion of correct responses that are unlikely to be due to chance. The relative decrease in accuracy here may be related to factors that also increase the RT, e.g., participants were less sure of the correct response so took slightly longer, and were slightly less likely to be correct. In this case, their correct responses may still be meaningful. For both slow and anticipatory responses, the number of trials drops significantly, and this alone could be used to achieve the same outcomes. Given this, I see no reason why simply removing RTs that deviate above a threshold (e.g., +/- 2.5 SDs from the mean) would not be equally or more effective, instead of using arbitrary bins. This is ultimately a small point, since not many trials were rejected overall, but the logic of using accuracy changes and binning in this way should be carefully considered.

3) Page 14, lines 42/43: Do the authors have any specific considerations regarding the differences in equipment from 45 years ago, with particular regard to how they may have influenced response times?

4) Figure 1: The colours for compatible and incompatible are reversed in the graph, relative to the figure legend. I suggest reversing the colours in the graph, as compatible is more intuitively associated with green, in which case the figure legend text is correct. Please ensure that the lines and the graph-legend “Cue Type” are changed.

Review form: Reviewer 3

Do you have any ethical concerns with this paper?

No

Have you any concerns about statistical analyses in this paper?

No

Recommendation?

Accept as is

Comments to the Author(s)

No comments

Decision letter (RSOS-190134.R2)

04-Oct-2019

Dear Mr McCormick

On behalf of the Editor, I am pleased to inform you that your Stage 2 Replication submission RSOS-190134.R2 entitled "On the Endogenous and Exogenous Selection of Signals" has been accepted for publication in Royal Society Open Science subject to minor revision in accordance with the referee suggestions. Please find the referees' comments at the end of this email.

The reviewers and Subject Editor have recommended publication, but also suggest some minor revisions to your manuscript. Therefore, I invite you to respond to the comments and revise your manuscript.

Please also ensure that all the below editorial sections are included where appropriate (a non-exhaustive example is included in an attachment):

- Ethics statement

- Data accessibility

<http://datadryad.org/submit?journalID=RSOS&manu=RSOS-190134.R2>

- Competing interests

- Authors' contributions

- Acknowledgements

- Funding statement

Because the schedule for publication is very tight, it is a condition of publication that you submit the revised version of your manuscript within 7 days (i.e. by the 12-Oct-2019). If you do not think you will be able to meet this date please let me know immediately.

- 1) A text file of the manuscript (tex, txt, rtf, docx or doc), references, tables (including captions) and figure captions. Do not upload a PDF as your "Main Document".
- 2) A separate electronic file of each figure (EPS or print-quality PDF preferred (either format should be produced directly from original creation package), or original software format)
- 3) Included a 100 word media summary of your paper when requested at submission. Please ensure you have entered correct contact details (email, institution and telephone) in your user account
- 4) Included the raw data to support the claims made in your paper. You can either include your data as electronic supplementary material or upload to a repository and include the relevant DOI within your manuscript
- 5) Included your supplementary files in a format you are happy with (no line numbers, Vancouver referencing, track changes removed etc) as these files will NOT be edited in production

Kind regards,
Andrew Dunn

on behalf of Professor Chris Chambers (Registered Reports Editor, Royal Society Open Science)
 openscience@royalsociety.org

Associate Editor Comments to Author (Professor Chris Chambers):

The Stage 2 manuscript was returned to the three original reviewers who assessed it at Stage 1. All are positive about the submission, noting that the Stage 1 primary criteria are met, but Reviewers 1 and 2 ask for clarification of the interpretation and Reviewer 1 queries the precision and validity of the data processing procedures. Once these minor issues are addressed, full acceptance should be forthcoming without requiring further in-depth review.

Reviewer: 1

Comments to the Author(s)

Minor:

I found the explanation for the discrepancy between the current results and those of Lawrence & Klein (2013) shown in fig 5 a bit hard to follow. As I understand it, L&K show faster RT but (slightly) reduced accuracy for a combined cue, whereas as the current study shows faster RT and improved accuracy. However, this pattern is also observed in the 'no signal' trials. There is some reference to participants having different mental states in the two studies. Is the idea that something about the design of the current study means that participants are better in the combined condition, irrespective of the presence of an alerting signal, and this is the cause of the inconsistency with L&K?

Final paragraph, 2nd sentence reads "This reinforces the theory that alerting increases response speed without improving the quality of information processed about the stimulus of interest, and instead is consistent with the suggestion that alerting changes the point at which information is consulted to generate a decision."

This is a bit confusing and the 1st & 2nd part of the sentence seem to be offering similar interpretations of the data, but the 'and instead' connector suggests the data are consistent with one interpretation but not the other

Reviewer: 2

Comments to the Author(s)

Thank you for your Stage 2 Submission; it was a pleasure to go through your results. Overall I am satisfied that this submission fulfills all the relevant criteria. Nonetheless, I do have some small points that would need to be addressed, or at the very least considered. Assuming that point 4 is addressed, I would be happy to accept this without further review (although I would prefer to see points 1 and 3 addressed as well).

1) Page 11, lines 42/43: The authors report a cut-off for trial exclusion as including "a respectable number of trials that fall within that bin". What is meant by respectable? Was there an objective threshold for this?

2) Page 11, line 36 to page 12, line 27 ("Data Processing" section): This binning method is in a way

refreshing, as most studies just use 'default' threshold of around 200 and 1800. However, I am not convinced that using changes in accuracy from bin to bin is a logically sound method for trial exclusion. For slow responses, the accuracy (84%) is still very high, suggesting a large proportion of correct responses that are unlikely to be due to chance. The relative decrease in accuracy here may be related to factors that also increase the RT, e.g., participants were less sure of the correct response so took slightly longer, and were slightly less likely to be correct. In this case, their correct responses may still be meaningful. For both slow and anticipatory responses, the number of trials drops significantly, and this alone could be used to achieve the same outcomes. Given this, I see no reason why simply removing RTs that deviate above a threshold (e.g., +/- 2.5 SDs from the mean) would not be equally or more effective, instead of using arbitrary bins. This is ultimately a small point, since not many trials were rejected overall, but the logic of using accuracy changes and binning in this way should be carefully considered.

3) Page 14, lines 42/43: Do the authors have any specific considerations regarding the differences in equipment from 45 years ago, with particular regard to how they may have influenced response times?

4) Figure 1: The colours for compatible and incompatible are reversed in the graph, relative to the figure legend. I suggest reversing the colours in the graph, as compatible is more intuitively associated with green, in which case the figure legend text is correct. Please ensure that the lines and the graph-legend "Cue Type" are changed.

Reviewers' comments to Author:

Reviewer: 3

Comments to the Author(s)

No comments

Author's Response to Decision Letter for (RSOS-190134.R2)

See Appendix C.

Decision letter (RSOS-190134.R3)

28-Oct-2019

Dear Mr McCormick:

It is a pleasure to accept your Stage 2 Replication entitled "On the Endogenous and Exogenous Selection of Signals" in its current form for publication in Royal Society Open Science.

Kind regards,
Lianne Parkhouse
Editorial Coordinator
Royal Society Open Science
openscience@royalsociety.org

on behalf of Professor Chris Chambers (Subject Editor)
openscience@royalsociety.org

Appendix A

To Dr. Chambers,

Thank you for your response. We were delighted to hear that you thought this manuscript was suitable for in-principal acceptance. We have addressed reviewer comments in-line below, and believe we are returning an improved version of our original submission because of the feedback. Reviewer comments are italicized, while our responses are bolded.

Reviewer: 1

My main concerns relate to the power and analysis pipeline

- 1. There is no discussion of how or why a sample of 24 participants was selected. I has expected to see a an apriori power calculation based on effects previously observed in the literature used to justify the sample size*

A sample of 24 in each signaling condition satisfies Simonsohn's (2015) criterion of 2.5 times the original sample size (Posner et al. N=9) to detect an effect the same size or larger. This was also clarified in a footnote. In addition, we are happy to provide a post-hoc measurement of precision (95% confidence intervals) in our analyses.

Simonsohn, U. (2015). Small telescopes: Detectability and the evaluation of replication results. *Psychological science*, 26(5), 559-569.

- 2. The analysis doesn't seem to address the hypothesis 2, which is " comparing performance in the high-intensity signal condition to the no-intensity signal condition to see if there are differences in the relationship between RT and ER, as predicted by Lawrence and Klein (2013)".*

The 2x2 anova described in the analysis section does not include a hgh v low intensity factor, so the these conditions are not statistically compared. It seems to me that a 2x2x2 omnibus anova would be more appropriate to address hypothesis 2.

We originally proposed *two* 4 (foreperiod) x 2 (compatibility) ANOVAs to address our hypotheses, but it is evident that a 4 (foreperiod) x 2 (compatibility) x 2 (intensity) omnibus ANOVA would be more appropriate. Thank you for this suggestion.

Reviewer: 2

Comments to the Author(s)

This study implements implements a relatively recent auditory stimulus technique, which accounts for differences between exogenous and endogenous alerting, and applies it to a well-established paradigm with the intention of replicating the results originally obtained from this paradigm, as well as determining the independent influence of endogenous and exogenous alerting on this established pattern of results.

Generally, the motivations of this study are well structured and clearly laid out, and without a doubt demonstrate the importance of carrying out this research. Most importantly, the description of the upcoming methods and analyses are put forward in a transparent fashion, though there are some discrepancies in the description of stimuli and procedures; hopefully these are due simply to errors in writing. Overall, several issues will need to be addressed before I can recommend this manuscript for publication in principle. These issues are laid out below:

- 1) When describing Lawrence & Klein's (2013) methodology, the description of the concept of "contingency" is lacking in clarification. Since I have read this paper myself, I understand that contingency in this case means that the order of presentation of signal and target was predictable and alternating. Nowhere in your manuscript is this clarified, and readers may be confused and potentially assume that you are referring to the identity or the location of the signal with respect to the target. Also, clarifying the actual procedure of Lawrence & Klein's experiment will help readers*

to understand that their task was qualitatively different, and to appreciate the motivation for the current study.

We have made our explanation of these methods clearer, as we agree that they are complex and important to understand.

2) The figures you have uploaded are extremely low in DPI quality, to the extent that they are difficult to read. Please increase the DPI substantially before re-uploading.

We have fixed our figures so they are now vector format.

3) On the 7th page (counting from the first page of the introduction), on line 17, the authors refer to "Figure 1", but are in fact referring to Figure 3.

Thank you for pointing this out.

4) On the 8th page, line 31/32, the authors state that the auditory signal was 50ms. However, in the footnote on the same page, they state that this signal was increased to 100ms. Please ensure this is consistent and correct.

This has been addressed.

5) On the 9th page, line 3, the authors state that the target presentation followed the end of the auditory signal by 100/250/850ms. However, these are SOA times (stimulus onset asynchrony), and the reason for the change in SOA compared to the original study was to ensure that the time between the *end* of the auditory signal and the onset of the target was in fact identical to the previous study. Therefore, I

strongly suspect that the authors actually mean to say that the time between the *start* of the auditory signal and the onset of the target was 100/250/850ms. Please revise this section and ensure all descriptions of stimulus timing are correct.

This has been clarified.

6) It is not clear why the SOA time needed to be increased simply because the auditory signal length increased; there is nothing special about the time between the end of the auditory signal and the start of the target onset; participants can begin to respond to the auditory stimulus as soon as it is perceived, which is why stimulus onset asynchrony is considered a more important metric. In my opinion, it would be better to keep the original SOA times unchanged.

When designing this replication, we elected to use a subset of the six SOAs to obtain more data per-participant in each condition. The no-signal condition was required, so this left five other options. We wanted to observe the U-function from the original study, so we elected to choose the shortest (50ms), middle (200ms), and longest (800ms) SOAs. Because our signal was 100ms instead of the original 50ms, we were required to double the time length of the 50ms SOA condition to ensure that the signal did not overlap with the presentation of the target. Then, to keep equal distance between the three signal SOAs, we elected to add 50ms onto the other two SOAs, resulting in our proposed 100ms, 250ms, and 850ms SOAs. This has been elaborated in footnote three.

7) The use of past and future tense throughout the methods section is inconsistent, e.g., “participants will be run...” vs. “participants completed four blocks...”. Please ensure you comply with the publication guidelines and keep the writing to past tense at this stage.

This has been fixed.

8) Finally, there are a very large number of writing errors and typos (e.g., the word "stimuli" is used for singular, when it should only be used for plural; "stimulus" is the singular term), as well as formatting errors (e.g., 5th page contains a sentence in a different typeface). These errors appear throughout, but are especially severe in the Figure legend of Figure 3. The authors should carefully review all sections of the manuscript to ensure no spelling or grammatical errors.

Provided the authors substantially reduce their grammatical/spelling errors, and add some additional clarifications in their introduction (particularly regarding the SOA variations relative to the original Posner study), I will be happy to accept this manuscript for publication in principle.

The article was searched for these errors and were corrected accordingly.

Reviewer: 3

Comments to the Author(s)

This study was designed to replicate the seminal study by Posner et al, 1973, with both endogenous and exogenous signals adopted in Lawrence and Klein, 2013. The authors, in doing this way, tried to explore the difference on speed-accuracy tradeoff between endogenous and exogenous signals.

The logic behind this study is very straightforward. This kind of comparison between endogenous and exogenous signals would have broad audiences in the field of attention. It also has strong general interesting. I only have some suggestions on the experimental design that might help improving the paper and make it more accessible to a wider audience.

*I, Based on the definition of different alerting signals in Lawrence and Klein (2013), and the logic behind the current study, I would like to see a fully design, i.e., 2 (Contingency: yes vs. no) * 2 (Intensity: change vs. no-change) * 4 (SOAs: 0, 50, 200, and 800 ms) * 2 (Compatibility: yes vs. no). I believed that, only in this way, the authors could compare endogenous (0Δ dB and C) and exogenous signals (+Δ dB and NC).*

Otherwise, as described in the paper that there was no NC condition, we could only examine the difference between endogenous signal (0Δ dB and C) and both endogenous and exogenous signal (+Δ dB and C).

We appreciate these suggestions. Where we are only interested in conditions in which the signals are informative of the presentation of a target, we will stick to our original design that compares the combined signaling condition to the purely endogenous condition. This allows us to replicate the methods of the original study (combined signal) and compare this to a novel condition (purely endogenous). Additionally, based on these comments, we improved the clarity of our definition of the combined signal condition.

2, As illustrated in Figure 1, 200 ms is the peak for RT, and 100 ms is the peak for accuracy. I think it is valuable to involve both in.

We have expanded upon our footnote justifying the subset of the original studies ‘foreperiods’. We wanted to still be able to observe the U-shaped functions, but also wanted to reduce the number of conditions so that each participant contributed more data to each combination. 100 and 200 were close enough in interval that we just chose one of the two.

Minor points

1, See p. 9. In the main text, the authors mentioned Figure 1 and 1a, but I cannot find them in Figure 1.

Please check it. (Same as R2)

2, Figure 2 is hard to be understood. Please give more explanation, or change it.

3, In Figure 3, I think it would be helpful for reader to understand, if you also put SOA 0 ms on it.

4, Figure 3 caption, line 39. ‘A interval’ – ‘An interval’.

We have made these changes. Thank you for the specific notes.

We appreciate the efforts of all individuals involved in this review process. We look forward to hearing back from you.

Sincerely,

Colin McCormick

Graduate Student

Dalhousie University

Appendix B

Professor Chris Chambers,

We thank you for your stage one acceptance, and excitedly present you with our stage two submission. We believe that we have followed all instructions indicated in the previous cover letter. You will find attached all appropriate documents.

Thanks,

Colin R. McCormick
Dalhousie University

Appendix C

Dr. Chambers,

We are very happy to hear that our stage two submission has been accepted with minor revisions. We thank the reviewers for their feedback. We have responded to the comments below in-line, and have indicated where changes have been made to the manuscript.

Associate Editor Comments to Author (Professor Chris Chambers):

The Stage 2 manuscript was returned to the three original reviewers who assessed it at Stage 1. All are positive about the submission, noting that the Stage 1 primary criteria are met, but Reviewers 1 and 2 ask for clarification of the interpretation and Reviewer 1 queries the precision and validity of the data processing procedures. Once these minor issues are addressed, full acceptance should be forthcoming without requiring further in-depth review.

Reviewer: 1

Comments to the Author(s)

Minor:

I found the explanation for the discrepancy between the current results and those of Lawrence & Klein (2013) shown in fig 5 a bit hard to follow. As I understand it, L&K show faster RT but (slightly) reduced accuracy for a combined cue, whereas as the current study shows faster RT and improved accuracy. However, this pattern is also observed in the 'no signal' trials. There is some reference to participants having different mental states in the two studies. Is the idea that something about the design of the current study means that participants are better in the combined condition, irrespective of the presence of an alerting signal, and this is the cause of the inconsistency with L&K?

The first part of this summary is correct: our signaling conditions produced different patterns of speed-accuracy performance in comparison to Lawrence and Klein. Separate from these phasic alerting effects, we also observed tonic effects of these signals in the trials in which no signal was presented. For the no-signal trials, we are theorizing that exposure to these two different signaling types across a block of trials is changing the overall mental state of the participant. So, the 'signaling conditions' do influence performance on both signal present and 'signal absent' trials, but this an independent observation that does not directly relate to the comparison between our signaling trial results and L&K's signaling trial results. We have modified the paragraph that follows figure 5 to make it more clear that this is independent of the phasic alerting comparisons.

Final paragraph, 2nd sentence reads "This reinforces the theory that alerting increases response speed without improving the quality of information processed about the stimulus of interest, and instead is consistent with the suggestion that alerting changes the point at which information is consulted to generate a decision."

This is a bit confusing and the 1st & 2nd part of the sentence seem to be offering similar interpretations of the data, but the 'and instead' connector suggests the data are consistent with one interpretation but not the other

Thank you for noticing this, we have fixed this part so that these two statements fit with one another.

Reviewer: 2

Comments to the Author(s)

Thank you for your Stage 2 Submission; it was a pleasure to go through your results. Overall I am satisfied that this submission fulfills all the relevant criteria. Nonetheless, I do have some small points

that would need to be addressed, or at the very least considered. Assuming that point 4 is addressed, I would be happy to accept this without further review (although I would prefer to see points 1 and 3 addressed as well).

- 1) Page 11, lines 42/43: The authors report a cut-off for trial exclusion as including “a respectable number of trials that fall within that bin”. What is meant by respectable? Was there an objective threshold for this?

Respectable referred to a sufficient number of trials being present within that bin to be able to make a judgement for accuracy. For instance, our 1-50ms bin of response times had an accuracy of 100%, but it was made up of only 6 trials, so we can't trust that these are not anticipatory responses despite the accuracy meeting our criteria. We have changed the use of the word respectable to increase clarity for this section.

- 2) Page 11, line 36 to page 12, line 27 (“Data Processing” section): This binning method is in a way refreshing, as most studies just use ‘default’ threshold of around 200 and 1800. However, I am not convinced that using changes in accuracy from bin to bin is a logically sound method for trial exclusion. For slow responses, the accuracy (84%) is still very high, suggesting a large proportion of correct responses that are unlikely to be due to chance. The relative decrease in accuracy here may be related to factors that also increase the RT, e.g., participants were less sure of the correct response so took slightly longer, and were slightly less likely to be correct. In this case, their correct responses may still be meaningful. For both slow and anticipatory responses, the number of trials drops significantly, and this alone could be used to achieve the same outcomes. Given this, I see no reason why simply removing RTs that deviate above a threshold (e.g., +/- 2.5 SDs from the mean) would not be equally or more effective, instead of using arbitrary bins. This is ultimately a small point, since not many trials were rejected overall, but the logic of using accuracy changes and binning in this way should be carefully considered.

The slower cut-off is to try to eliminate trials in which participants miss the onset of the stimulus, likely due to a lapse in vigilance. The bulk of modern RT modelling recognizes either a fall off or dip in accuracy at late RTs. Although we agree that 84% is a relatively high accuracy rate, this reflected a dip of 6% after having accuracy in the low 90's consistently across the RT bins leading up to that point. This is in addition to the frequency being cut in half (185 to 94 trials), which, as this reviewer points out, also likely predicts a change in task-related behaviour. Whereas we are very likely getting rid of some real response data, we believe that the general procedure is logical for cutting down noise, and is certainly less arbitrary than picking some SD threshold or predetermined range of RTs. This procedure has been used in past published research (see below citation). In addition to these points, we also think it would be more than awkward to change methods of analysis from those approved as part of the registration process.

Christie, J., Hilchey, M. D., Mishra, R., & Klein, R. M. (2015). Eye movements are primed toward the center of multiple stimuli even when the interstimulus distances are too large to generate saccade averaging. Experimental Brain Research, 233(5), 1541-1549. doi: 10.1007/s00221-015-4227-7

- 3) Page 14, lines 42/43: Do the authors have any specific considerations regarding the differences in equipment from 45 years ago, with particular regard to how they may have influenced response times?

As we do not have an explicit considerations for equipment, we have instead changed this section to mention that the differences in speed could be related to a few possible changes in the 45 year gap, with equipment and cohort effects being possible options.

- 4) Figure 1: The colours for compatible and incompatible are reversed in the graph, relative to the figure legend. I suggest reversing the colours in the graph, as compatible is more intuitively associated with

green, in which case the figure legend text is correct. Please ensure that the lines and the graph-legend "Cue Type" are changed.

We have fixed this to be more logical as suggested.

Reviewers' comments to Author:

Reviewer: 3

Comments to the Author(s)

No comments

—

Please let us know if there is anything else that is required on our end.

Sincerely,

Colin R. McCormick